# l-Quebrachitol Promotes the Proliferation, Differentiation, and Mineralization of MC3T3-E1 Cells: Involvement of the BMP-2/Runx2/MAPK/Wnt/β-Catenin Signaling Pathway

**DOI:** 10.3390/molecules23123086

**Published:** 2018-11-26

**Authors:** Thanintorn Yodthong, Ureporn Kedjarune-Leggat, Carl Smythe, Rapepun Wititsuwannakul, Thanawat Pitakpornpreecha

**Affiliations:** 1Department of Biochemistry, Faculty of Science, Prince of Songkla University, Hat-Yai, Songkhla 90110, Thailand; thanintorn.y@gmail.com; 2Department of Oral biology and Occlusion, Faculty of Dentistry, Prince of Songkla University, Hat-Yai, Songkhla 90110, Thailand; ureporn.l@psu.ac.th; 3Department of Biomedical Science, University of Sheffield, Sheffield, England S10 2TN, UK; c.g.w.smythe@sheffield.ac.uk; 4Center of Excellence in Natural Rubber Latex Biotechnology Research and Development, Prince of Songkla University, Hat-Yai, Songkhla 90110, Thailand; wrapepun@yahoo.com

**Keywords:** l-quebrachitol, Osteoblastogenesis, Wnt/β-Catenin, BMP-2, Runx2, MAPK

## Abstract

Osteoporosis is widely recognized as a major health problem caused by an inappropriate rate of bone resorption compared to bone formation. Previously we showed that d-pinitol inhibits osteoclastogenesis but has no effect on osteoblastogenesis. However, the effect on osteoblast differentiation of its isomer, l-quebrachitol, has not yet been reported. The purpose of this study was, therefore, to investigate whether l-quebrachitol promotes the osteoblastogenesis of pre-osteoblastic MC3T3-E1 cells. Moreover, the molecular mechanism of action of l-quebrachitol was further explored. Here, it is shown for the first time that l-quebrachitol significantly promotes proliferation and cell DNA synthesis. It also enhances mineralization accompanied by increases in mRNA expression of bone matrix proteins including alkaline phosphatase (ALP), collagen type I (ColI), osteocalcin (OCN), and osteopontin (OPN). In addition, l-quebrachitol upregulates the mRNA and protein expression of bone morphogenetic protein-2 (BMP-2) and runt-related transcription factor-2 (Runx2), while down-regulating the receptor activator of the nuclear factor-*κ*B ligand (RANKL) mRNA level. Moreover, the expression of regulatory genes associated with the mitogen-activated protein kinase (MAPK) and wingless-type MMTV integration site (Wnt)/β-catenin signaling pathways are also upregulated. These findings indicate that l-quebrachitol may promote osteoblastogenesis by triggering the BMP-2-response as well as the Runx2, MAPK, and Wnt/β-catenin signaling pathway.

## 1. Introduction

Osteoporosis is a highly prevalent bone disease that results from inappropriate bone remodeling caused by an imbalance between osteoclastic bone degradation and osteoblastic bone regeneration. The condition is characterized by decreased bone mass and degeneration of bone architecture, which results in decreased bone strength and increased susceptibility to fractures. The World Health Organization (WHO) considers osteoporosis to be a major health problem as it affects over 200 million people worldwide [1]. Age-related bone loss, which affects approximately 50% of women and 25% of men over 50 years old, is the most important lifetime risk factor for having an osteoporosis-related fracture [2].

The strategy for treatment of osteoporosis is to enhance bone formation through an induction of osteoblastogenesis by increasing osteoblast proliferation and differentiation with an anabolic agent such as estrogen, and/or the reduction of osteoclastogenesis with anti-resorptive drugs, such as a bisphosphonate. However, despite being effective, treatment with estrogen and bisphosphonate is associated with serious side effects, such as ovarian carcinogenesis and bone quality reduction. Therefore, research has focused on identifying suitable diets or supplements, particularly those derived from natural resources to prevent osteoporosis [3].

Multiple signaling pathways, including runt-related transcription factor 2 (Runx2), osterix (Osx), bone morphogenic protein (BMP), mitogen-activated protein kinase (MAPK), and the wingless-type MMTV integration site (Wnt)/β-catenin pathways, regulate the proliferation and differentiation of osteoblasts and thus are involved in controlling bone formation. In addition, the increased expression of bone matrix proteins, such as alkaline phosphatase (ALP), type I collagen (ColI), and osteopontin (OPN) also stimulate mineralization and lead to bone formation [4]. Osteoblasts not only regulate bone formation but also control bone resorption by modulating osteoclastogenesis via osteoprotegerin (OPG)/receptor activator of nuclear factor-κB ligand (RANKL)/ receptor activator of nuclear factor-κB (RANK) system [4].

Many plant-based compounds have been found to promote osteoblastogenesis-induced bone formation, including glycosides, flavonoids, terpenoids, coumarins, phenols, phenolics acids, and others. Moreover, molecular mechanisms and various cellular targets have been identified, including the control of transcription factors, bone-specific protein, signal pathways, and the OPG/RANKL system [3]. Among compounds derived from natural resources, some simple carbohydrates are also potential candidates as an alternative treatment or form of prevention against osteoporosis. However, their anti-osteoporosis pharmacological mechanism is basically related to the reduction of bone resorption. Both polymeric and monomeric sugars have shown promising activity on the suppression of osteoclastogenesis. For example, β-Glucans, polymers of d-glucose linked via β-(1,3), β-(1,4), or β-(1,6)-glycosidic bonds, from the mushroom, *Pleurotus citrinopileatus*, inhibit osteoclast differentiation and activity through inhibition of RANKL and tartrate-resistant acid phosphatase, respectively. In addition, β-glucans from the fungus, *Aureobasidium pullulans*, exhibit effects on bone metabolism both in vivo and in vitro. In ovariectomized rats, an animal model for postmenopausal osteoporosis [5], β-glucans suppresses bone loss while the exposure of pre-osteoblastic MC3T3-E1 cells to β-glucans enhances their proliferation and mineralization [6]. 

Osteoclast differentiation is also regulated by various monomeric sugars via different molecular mechanisms. High glucose concentrations inhibit osteoclastogenesis through the suppression of RANKL-induced generation of reactive species (ROS) [7] and the reduced expression of several genes involved in differentiation, such as NFATc1 [8]. The rare aldohexose, allose, inhibits osteoclast differentiation via the strong induction of thioredoxin-interacting protein (TXNIP) which, in turn, modulates the thioredoxin-regulated gene transcription [9]. Recently, the amino sugars glucosamine (GLcN) and its acetyl derivative, *N*-acetyl glucosamine (GLcNAc), have been shown to suppress osteoclast differentiation through the promotion of *O*-GlcNAcylation [10,11]. GLcN not only suppresses osteoclastogenesis but also promotes osteoblastogenesis and increases autophagy by inhibiting the mammalian target of the rapamycin (mTOR) pathway [12]. Moreover, the intake of GLcN reduces bone loss in ovariectomized mice by suppressing osteoclast function [13]. The intracellular sugar, inositol and its derivatives, which act as essential messenger molecules in signal transduction, have recently been demonstrated to affect bone metabolism. Inositol hexakisphosphate (IP6), the major intracellular phosphorylated form of inositol, has been reported not only to block the mineralization of MC3T3-E1 osteoblasts, but also have a selective inhibitory effect on osteoclast formation [14]. The inhibition of osteoblast mineralization is brought about by neutralizing crystal growth and promoting the expression of the mineralization inhibitor osteopontin, without impairment of other specific bone matrix proteins [15]. Ca-Mg-IP6 consumption has been shown to bring about a reduction in bone loss of ovariectomized rats [16]. d-pinitol (3-*O*-methyl-d-chiro-inositol), is one of the natural occurring derivatives of inositol. It has been reported to suppress RANKL-induced osteoclastogenesis and protect bone loss in the ovariectomy animal model but has no effect on the proliferation and differentiation of osteoblast cells [17].

l-quebrachitol (2-*O*-methyl-l-chiro-inositol) is a naturally-occurring, optically-active methoxy analog of inositol (Figure 1). In addition to the many plants that are reported to contain this compound, the serum of rubber latex has a particularly high concentration, suggesting that l-quebrachitol could be recovered as a byproduct from the rubber industry. l-quebrachitol has acquired increasing interest as a starting material in the chiral synthesis of various biologically active compounds. In addition, it has been reported to have various biological activities, including free-radical scavenging, gastroprotection, anti-platelet aggregation, and anti-diabetic activity [18]. Its isomer, d-pinitol (Figure 1), possesses inhibitory activity against osteoclastogenesis without any effect on osteoblastogenesis [17]. Although they have the same chemical structures, most isomers demonstrate marked differences in biological activities such as pharmacology, toxicology, metabolism, and pharmacokinetics [19]. Because l-quebrachitol has not yet been investigated for any effect on either osteoclastogenesis or osteoblastogenesis, the aim of this study is to investigate the effect of l-quebrachitol on the proliferation and differentiation of the murine pre-osteoblastic cell line MC3T3-E1 and explore the molecular mechanism by which l-quebrachitol promotes osteoblastogenesis. In addition, we also explore a method to obtain the purified l-quebrachitol as a by-product from the serum obtained along with the rubber sheeting process.

## 2. Results

### 2.1. Extraction and Purification of l-quebrachitol from Rubber Latex Serum

Our study demonstrated a method for the recovery and purification of l-quebrachitol as a by-product from the rubber sheeting process. Briefly, fresh latex (7 L) was coagulated by addition of formic acid, squeezed to obtain 3.5 L of the serum, which was further fractionated by molecular size-dependent differential filtration. The permeate arising from the use of a 10 kDa filter membrane was further fractioned using a 1 kDa membrane. The resultant permeate (3.5 L) was concentrated by spray-drying to obtain 100 g of dried powder, then extracted with ethanol, before ionic impurities were removed by cation-exchange chromatography. The purified l-quebrachitol was characterized analytically by comparison of chromatographic and spectroscopic data, including mass, IR, and NMR spectra with a commercial standard preparation (Sigma-Aldrich, Missouri). The yield (*means ± SEM*, *n* = 5) of l-quebrachitol was 0.92 ± 0.22 g/L of fresh latex or 1.85 ± 0.45 g/L of serum. 

### 2.2. l-quebrachitol Enhances the Cell Viability of Pre-Osteoblastic MC3T3-E1 Cells

Using an MTT assay which measures metabolic activity as a surrogate indicator of cell viability, cells were exposed to a range of l-quebrachitol concentrations for 24–72 h. The results obtained at 24 and 48 h after exposure indicated that l-quebrachitol is not only not cytotoxic to pre-osteoblastic MC3T3-E1 cells at concentrations ranging from 0.001 to 1000 µg/mL, but also that it significantly increased cell proliferation in a broadly concentration-dependent manner from 0.01 to 100 µg/mL. However, l-quebrachitol-mediated cell proliferation decreased to control levels by 72 h, perhaps reflecting a limiting effect of cell density on proliferation. At very high concentrations (1000 µg/mL), l-quebrachitol showed some toxicity towards these cells, reducing cell numbers by about 20% after 72 h of exposure (Figure 2).

### 2.3. l-quebrachitol Promotes Cell DNA Synthesis

To further understand the effects of L-quebrachitol on proliferation, we examined the effect on cell cycle progression, by the determination of the percentage of cells in each cell cycle phase (G0/G1, S, and G2/M) by flow cytometry after treatment with various concentrations of l-quebrachitol. The results indicated that the proportion of cells in the G0/G1 phase was significantly decreased after treatment with 0.001, 0.01, 0.1, 1, and 10 µg/mL of l-quebrachitol for 48 h, whereas the percentage of cells in S phase was significantly increased. These results suggest that progression into S-phase is promoted by l-quebrachitol (Figure 3A,B).

### 2.4. l-quebrachitol Promotes Differentiation and Mineralization of Osteoblast Cells

The pre-osteoblastic MC3T3-E1 cells were induced to differentiate with an osteogenic medium and the effect of l-quebrachitol on cell differentiation was evaluated. Matrix mineralization was assessed by visualizing the extent of the Alizarin Red S staining of cellular calcium deposits after cells were incubated with various concentrations of l-quebrachitol for 14 or 21 days. The amount of calcium deposit markedly increased with l-quebrachitol concentrations of 0.1 and 1 µg/mL by about 2 to 2.5 times, respectively, compared with the control at 14 days after treatment. However, even though the level of mineralization appeared to have decreased at 21 days compared to that at 14 days, l-quebrachitol concentrations of 0.001–1 µg/mL still augmented the amount of calcium deposit compared to the control at 21 days. Alizarin Red S staining observed in Figure 4A was quantified (Figure 4A,B), confirming the increase observed at 0.1–1 µg/mL after 14 days of exposure. In addition, alkaline phosphatase, the early-stage biomarker for matrix maturation [20], was also measured following treatment with l-quebrachitol for 7 days, demonstrating that l-quebrachitol concentrations of 0.01–1 µg/mL significantly increased the cellular ALP activity (Figure 4C). Taken together, these results indicate that l-quebrachitol significantly enhances the differentiation and matrix maturation of osteoblastic cells.

### 2.5. l-quebrachitol Up-Regulates mRNA Expression of BMP-2, Runx2, and Osteogenesis Markers

To determine the effect of l-quebrachitol on osteoblastic cell differentiation, the expression of genes involved in the formation of the cellular matrix including the early-stage (alkaline phosphatase), middle stage (collagen type 1 and osteopontin), late stage (osteocalcin), and key regulatory (Bone morphogenetic protein-2 and Runt-related transcription factor-2) genes [21] was evaluated by qRT-PCR after treatment with l-quebrachitol. The analysis revealed that l-quebrachitol significantly increased the expression of osteocalcin, osteopontin, runt-related transcription factor-2, and alkaline phosphatase genes after 24 h exposure, although the optimal concentration for maximal gene expression varied from gene to gene. In contrast, the expression of the *collagen type 1* gene seemed to slightly increase after 72 h of exposure. l-quebrachitol increased the expression of the bone morphogenetic protein-2 gene, which plays a crucial role in the transduction of osteoblastic differentiation and bone formation, with an expression level about 4-fold higher than control at concentrations of 0.01 and 10 µg/mL after 24 h. Consistent with the notion that l-quebrachitol can induce osteoblastic cell differentiation, all the tested concentrations of l-quebrachitol significantly suppressed the expression of RANKL, a major regulatory gene for osteoclastogenesis (Figure 5).

### 2.6. l-quebrachitol Increases Expression of a Gene Involved in the MAPK and Wnt/β-Catenin Signaling Pathways

To investigate the participation of signaling pathways mediating the effect of l-quebrachitol on osteoblast proliferation and differentiation, the expression of mitogen-activated protein kinase (MAPK) family genes, as well as the Wnt/β-catenin signaling pathway, were investigated with qRT-PCR. It has been demonstrated that MAPKs play an important role in controlling cell proliferation and differentiation and regulate osteoblast differentiation [22]. The expression of MAPK pathway members such as JNK1, JNK2, ERK1, ERK2, and p38α was investigated in response to l-quebrachitol. The results indicated that lower concentrations (0.01 and 0.1 µg/mL) of l-quebrachitol significantly induce ERK1 and ERK2 mRNA level both at 24 and 48 h, while the expression of JNK1 and JNK2 were also up-regulated with 0.01 and 0.1 µg/mL of l-quebrachitol only at 12 h. In addition, 0.01 µg/mL of l-quebrachitol significantly increased the mRNA level of p38α at all times tested (Figure 6). 

Wnt/β-catenin signaling has been demonstrated to regulate both osteoblast differentiation and bone formation [23]. Thus, the expression of genes related to Wnt/β-catenin signaling pathway regulators, including LRP5, β-catenin, Wnt5a, and Fzd4 was investigated with regard to the effect of l-quebrachitol. l-quebrachitol treatment tended to significantly up-regulate the mRNA expression of LRP5, β-catenin, and Fzd4 at 12 h, whereas the Wnt5a expression level increased and reached the highest level, about 2-fold, with 0.01 µg/mL of l-quebrachitol at 24 h. However, the mRNA expression of all genes tested was found to decrease at 48 h (Figure 7). Taken together, the l-quebrachitol treatment effectively induces the expression of genes related to Wnt/β-catenin signaling pathway regulators at 12–24 h, leading to the enhanced expression of osteoblast differentiation and proliferation markers.

### 2.7. Western Blot Analysis of BMP-2 and Runx2 Expression

To further verify that l-quebrachitol regulates components of the BMP signaling pathway and the expression of the master transcription factor, Runx2, we examined the protein expression of those molecules by Western blot analysis (Figure 8A). l-quebrachitol significantly enhanced the protein level expression of Runx2 at concentrations of 1 and 10 µg/mL at 10 days (Figure 8A), whereas BMP-2 protein expression significantly increased at 12 days with 1 and 10 µg/mL of l-quebrachitol and gave the highest level, 2-fold compared to the control, at 10 days with 0.01 µg/mL of l-quebrachitol (Figure 8B). 

## 3. Discussion

Previous studies demonstrated that *Hevea brasiliensis* latex serum is a potential source of a wide range of biologically-active constituents and enzymes [24,25,26,27,28], such as l-quebrachitol [29]. Thus, this study reported an alternative method for the isolation and purification of l-quebrachitol from *Hevea brasiliensis* latex serum with a recovery yield of 1.85 ± 0.45 g/L of serum, accompanied by chitinolytic enzymes [28] and a rubber sheet product (data not shown).

d-pinitol, an isomer of l-quebrachitol, has been reported to inhibit bone resorption through the suppression of RANKL-induced osteoclastogenesis with no effect on the proliferation and differentiation of osteoblasts [17]. However, the effect of l-quebrachitol on bone formation had not previously been studied. Hence, our study aimed to investigate its effect on osteoblastogenesis and the signaling pathway involved in the osteogenic activity of l-quebrachitol. Our results indicate that treatment with l-quebrachitol at concentrations of 0.01, 0.1, 1, 10, and 100 µg/mL significantly increased the cell viability of pre-osteoblastic MC3T3-E1 cells at 24 and 48 h (Figure 2). This is consistent with the results obtained from cell cycle analysis, which indicated that l-quebrachitol promotes the proliferation of cells by increasing cell DNA synthesis (Figure 3).

The BMP signaling pathway is a crucial signaling pathway in bone formation. BMP-2 reportedly acts as a potent inducer of a mature differentiation of pre-osteoblastic cells by regulating *Runx2*, a transcription factor that is required for bone formation. The activation of *Runx2* further regulates the expression of osteoblast differentiation by increasing alkaline phosphatase (ALP) activity and synthesis of the bone matrix proteins, including collagen type I (ColI), osteocalcin (OCN), and osteopontin (OPN), therefore, inducing bone formation and bone remodeling [3,30]. It has been demonstrated that *Runx2*-null mice have no bone tissue and the reduction in *Runx2* expression leads to a decrease in the bone matrix proteins [31]. In this study, l-quebrachitol significantly enhances the expression of BMP2 mRNA after either 24 or 48 h of exposure (Figure 5), consistent with the subsequent increase in BMP2 protein level at days 10 and 12 (Figure 8B). This is also consistent with an increase in the downstream regulator Runx2, at both mRNA (Figure 5) and protein expression levels (Figure 8A). In addition, the mRNA expression level of ALP, which is the most widely recognized marker of early-stage osteoblastic differentiation and an up-regulator of bone matrix genes, including ColI and OPN [32], is transiently increased at 24 h (Figure 5) consistent with the increase in ALP activity observed at 7 days (Figure 4C). Moreover, other bone matrix proteins, OCN, OPN, and ColI are also up-regulated at the mRNA level (Figure 5). Together, this leads to the enhancement of the late stage of osteoblastic differentiation, observed by the Alizarin Red S staining of the calcium deposit in mature osteoblasts at 14 days (Figure 4A,B). Hence, l-quebrachitol most likely promotes and up-regulates osteoblast maturation and differentiation from the early to the late stages through the BMP signaling pathway.

The canonical Wnt/β-catenin signaling pathway is one established pathway which plays a pivotal role in bone formation by modulating proliferation, differentiation, and mineralization in osteoblastogenesis. The Wnt/β-catenin signaling pathway is activated when extracellular Wnt ligands bind to frizzled receptors (Fzds) and its co-receptors LRP5 and LRP6, leading to a cascade of phosphorylation. This, in turn, results in destabilization of an intracellular protein complex facilitating a release of β-catenin which translocates into the nucleus, thus activating target gene expression mediated by TCF/LEF [33,34]. In addition, canonical Wnt/β-catenin signaling is active in various osteoblast or pre-osteoblastic cell lines such as MC3T3-E1 and cooperatively controls the osteoblast differentiation and bone formation via crosstalk with the BMP signaling pathway, suggesting that the Wnt/β-catenin signaling pathway is an upstream activator of BMP2 expression in osteoblasts [22]. Previously, it has been reported that the deactivation of Wnt/β-catenin signaling results in decreased OPG expression, increased RANKL expression, and the inhibition of osteoblastic differentiation [35]. Moreover, *LRP5* mutations cause changes in bone mass and deletion of *FZD9* is characterized in part by low bone density [33]. In this study, the qRT-PCR analysis showed that the expression level of *LRP5, β-catenin, Fzd4,* and *Wnt5a* significantly increased after treatment with l-quebrachitol (Figure 7). It was previously reported that the Wnt/β-catenin signaling pathway can induce osteoblast differentiation genes, such as *ALP, ColI,* and *OPN* [36], demonstrating that those genes are up-regulated following cell exposure to l-quebrachitol. These results suggest that the enhancement of osteoblast differentiation by l-quebrachitol is likely to correlate with the Wnt/β-catenin signaling pathway. 

MAPKs also play an important role in bone formation. MAPK family proteins, p38 mitogen-activated protein kinases (p38), extracellular signal-regulated kinases (ERK), and jun n-terminal kinase (JNK), are essential factors in the regulation of osteoblast differentiation through the activation of transcriptional factors, such as *AP-1* [37]. In addition, p38 activation seems to be critical for the regulation of ALP expression during MC3T3-E1 differentiation [38]. Remarkably, it has been reported that the MAPK pathway is able to control Runx2 and osterix via phosphorylation [39] and MAPK activation can induce Runx2-dependent osteocalcin and osteopontin genes [40]. In our study, l-quebrachitol was associated with the increased expression of the MAPK gene family members *JNK1*, *JNK2*, *ERK1*, *ERK2*, and *p38α* at the mRNA level (Figure 6), which suggests that l-quebrachitol may possibly promote osteoblast differentiation via the activation of the MAPK signaling pathway. 

Osteoblasts additionally control bone resorption by regulating osteoclastogenesis through osteoprotegerin (OPG)/receptor activator of nuclear factor-κB ligand (RANKL)/receptor activator of nuclear factor-*κ*B (RANK) system. RANKL plays a critical role in the activation of osteoclastogenesis via binding to its receptor RANK [4]. In our study, l-quebrachitol clearly suppresses the RANKL mRNA expression (Figure 5). These results suggest l-quebrachitol not only promotes bone formation through the induction of osteoblastogenesis, but also suppresses osteoclastogenesis through the OPG/RANKL/RANK system. 

In summary, we demonstrated for the first time that l-quebrachitol at a low concentration (0.01 µg/mL or 50 nM) can promote osteogenic proliferation, differentiation, and mineralization of pre-osteoblastic MC3T3-E1 cells accompanied by an increase in the expression of bone matrix proteins including ALP, ColI, OCN, and OPN. The osteoblastogenesis activity of l-quebrachitol is possibly mediated by triggering the BMP-2-responsive, Runx2, MAPK, and Wnt/β-catenin signaling pathways. Interestingly, the biological activity of l-quebrachitol to promote osteoblastogenesis contrasts starkly with its closely related structural isomer, d-pinitol, which promotes the suppression of osteoclastogenesis. Notably, this is the first report of a new signaling pathway involved in the promotion of osteoblastogenesis induced by monomeric polyols. Although further studies are required to study the effect on osteoclastogenesis and to clarify the in vivo activity and mechanisms, l-quebrachitol might be a candidate for a bone formation-promoting supplement. Moreover, this study could lead to value-creation as well as value-adding to natural rubber latex serum from rubber factories. 

## 4. Materials and Methods

### 4.1. Materials

Minimum essential medium alpha medium (α-MEM), RPMI-1640 phenol red-free medium and fetal bovine serum (FBS) were purchased from Gibco (Grand Island, NY, USA). β-Glycerophosphate, ascorbic acid, MTT (3-(4,5-dimethylthiazol-2-yl)-2,5-diphenyltetrazolium bromide), and cetylpyridinium chloride (CPC) were purchased from Sigma Chemical Co. (St. Louis, MO, USA). All other chemicals were of the grade quality available.

### 4.2. Collection and Fractionation of Fresh Latex

Fresh latex (7 L) was gathered in an ice-chilled beaker (Merck, Kenilworth, NJ, USA) from regularly tapped *H. brasiliensis* trees of the RRIM 600 clone. Latex coagulation was induced with the addition of 0.2% (*v*/*v*) aqueous formic acid. The rubber latex coagulum was squeezed through rollers to assemble the serum drainage. The collected serum, which contained 1.52 mg of the ML-1 protein, was then concentrated 20-fold by ultrafiltration with a Pillicon^®^ tangential flow ultrafiltration system (Millipore, Jaffrey, NH, USA) equipped with a 10 kDa cutoff membrane cassette and operated under 20 psi at room temperature. The 10 kDa permeate stream was further filtered with a smaller pore size (1 kDa cutoff) to obtain a 1 kDa permeate filtrate (3.5 L) and then concentrated with the mini spray dryer B-290 (BUCHI, New Castle, DE, USA) to collect the solid powder or 1 kDa permeate powder (100 g) and was used as a starting material for l-quebrachitol purification.

### 4.3. Purification of l-quebrachitol

Fifty grams of 1 kDa permeate powder was suspended in methanol at a ratio of 1:10 (*w*/*v*) and stirred at room temperature for 24 h. After this time, the insoluble components were removed by filtration with Whatman grade no. 1 filter paper (Fisher Scientific, Loughborough, Leicestershire, UK) and the filtrate obtained was concentrated at a 1/10 volume by a rotary evaporator (BUCHI, New Castle, DE, USA) at 40–45 °C. The resulting crystals were collected after 12 h. Additionally, they were recrystallized three times with 75% aqueous ethanol to give a colorless crystal. After freeze-drying (6.5 g), the samples were dissolved in double the amount of distilled water before being desalted through a cation-exchange column on AG 50W-X2 (Bio-Rad, Hercules, CA, USA) (5 × 10 cm, H^+^ form). The unbound fraction was washed with distilled water (300 mL), concentrated again by the rotary evaporator (BUCHI, New Castle, DE, USA), and then precipitated with cold ethanol (three volumes) at 4 °C overnight. The collected pellet, pure l-quebrachitol, was freeze-dried before further characterization (yield of 6.48 ± 1.57 % *w*/*w* of 1 kDa-permeate powder). The properties of the standard l-quebrachitol (Sigma-Aldrich, St. Louis, MO, USA) were confirmed through a comparison of chromatographic and spectroscopic data which included mass (Mass Spectrometer 2690, Milford, MA, USA), IR (Equinox 55, Bruker, Germany), and NMR spectra (Fourier Transform NMR Spectrometer 500 MHz, Unity Inova, Varian, Germany).

### 4.4. Cell Culture and Differentiation

The murine pre-osteoblastic cell line MC3T3-E1 (subclone 14 CRL-2593) was purchased from ATCC (Manassas, VA, USA). MC3T3-E1 cells were used in passage number 23 in all experiments. Thes early passage cells had not changed in morphology and osteoblastic function due to the long culture time [41]. Cells were cultured in α-MEM with 10% FBS, 100 units/mL of penicillin, and 100 μg/mL of streptomycin at 37 °C in a 5% CO_2_ incubator (Thermo Fisher Scientific, Waltham, MA, USA). The culture medium was changed every 2–3 days. To induce differentiation, cells were cultured in osteogenic induction medium containing 1 M of β-glycerophosphate and 50 μg/mL of ascorbic acid.

### 4.5. Cytotoxicity Assay

The MTT assay was performed to assess the cytotoxicity of l-quebrachitol. Cells were treated with l-quebrachitol at different concentrations. After treatment at 24 and 48 h, cultured cells were washed with PBS. RPMI-1640 phenol red-free medium and 5 mg/mL of MTT were added, and the mixture was incubated at 37 °C for 3 h. The formazan product was solubilized in acidic isopropanol, and the absorbance was measured at 570 nm.

### 4.6. Alkaline Phosphatase (ALP) Activity Assay

Cells were treated with l-quebrachitol for 7 days. The lysate cell supernatant was collected for assay. Protein concentration was measured by the Pierce BCA Protein Assay Kit (Thermo Fisher Scientific, Waltham, MA, USA) and ALP activity was determined with a substrate containing 4 mg/mL of 4-nitrophenyl phosphate (4NPP) in 0.2 M of 2-amino-2-methyl-1-propanol with 4 mM of MgCl_2_ at 37 °C for 30 min. The reaction was stopped by 0.1 M NaOH, and the absorbance of the yellow solution was measured at 405 nm. The ALP activity was normalized to the protein concentration and the enzyme activity is shown as μmole/min/mg of protein.

### 4.7. Alizarin Red S Staining

Cells were treated with l-quebrachitol for 14 or 21 days. Calcium deposition was measured using Alizarin red S staining. Briefly, cells were washed with PBS and fixed with 10% formaldehyde for 15 min. Cells were stained with 40 mM of Alizarin red S solution (pH 4.1–4.3) in the shaker (Labnet International, Big flats, NY, USA) at room temperature for 30 min. The non-specific staining was removed by washing with distilled water five times (5 min/time). Alizarin red S staining was dissolved by cetylpyridinium chloride (CPC) for quantification and the absorbance was measured at 550 nm.

### 4.8. Cell Cycle Analysis

Cells treated with l-quebrachitol for 48 h were trypsinized, fixed in ice-cold 70% ethanol at 4 °C at least 30 min, and washed with PBS after fixing. Then, propidium iodide containing RNase staining solution (Merck, New Jersey, USA) was added to determine the DNA content and incubated for 30 min at room temperature in the dark. Each sample (1 × 10^6^ cells) was analyzed by flow cytometry (Guava easyCyte^TM^ HT, Hayward, CA, USA) to determine the percentages of G1, S, and G2 in the cell cycle.

### 4.9. Quantitative Real-Time Polymerase Chain Reaction (qRT-PCR)

The total RNA from cells treated with l-quebrachitol was extracted using the TriPure Isolation Reagent (Roche, Buonas, Switzerland). RNA (1 μg) was reverse transcribed using the Transcriptor First Strand cDNA Synthesis Kit (Roche, Buonas, Switzerland) to cDNA according to the manufacturer’s instruction. The qRT-PCR analysis was performed using the EvaGreen HRM Mix (Solis Biodyne, Tartu, Estonia), and the specific primers for the genes are listed in Table 1. GAPDH was used as an internal control. The relative expression ratio was calculated using the 2^−△△Ct^ method.

### 4.10. Western Blot Analysis

MC3T3-E1 cells treated with l-quebrachitol for the indicated time were lysed with lysis buffer, and the concentration of protein was measured with a protein assay kit (Bio-Rad, Hercules, CA, USA). For Western blotting, each sample was separated on a 10% SDS-PAGE gel, followed by transfer to nitrocellulose membranes (Amersham Pharmacia Biotech, Amersham, Buckinghamshire, UK) by electroblotting. The membranes were blocked for nonspecific binding in 5% non-fat dried milk in TBS for 1h. BMP-2 and Runx2 expressions were detected by incubation with their specific primary antibodies, anti-BMP-2 (Abcam, Milton, UK) and anti-Runx2 (Cell Signaling Technology, Beverly, MA, USA) for 2 h. After washing with TBS-Tween 3 times, specific protein bands were probed with an Alexa infrared dye-conjugated secondary antibody (Invitrogen, Carlsbad, CA, USA) and visualized by the Odyssey Infrared Imaging System (LI-CORE) in accordance with the manufacturer’s instructions. 

### 4.11. Statistical Analysis

Data were presented as the mean ± standard error of the mean (SEM). All the results of the experiments were analyzed with the SPSS 23 statistical software (SPSS Inc., Chicago, IL, USA). The significance was analyzed by one-way analysis of variance (ANOVA), followed by Duncan′s multiple range test. *p* values < 0.05 were considered statistically significant.

## 5. Conclusions

In conclusion, we reported, for the first time, that l-quebrachitol promotes osteoblast proliferation, differentiation, and mineralization, possibly through the involvement of the BMP, MAPK, and Wnt/β-catenin signaling pathways. In addition, a method to obtain purified l-quebrachitol as a byproduct from *Hevea* latex serum was reported.

## Figures and Tables

**Figure 1 molecules-23-03086-f001:**
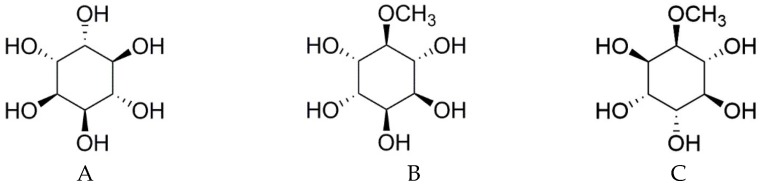
The structures of d-chiro-inositol (**A**), d-pinitol (**B**) and l-quebrachitol (**C**).

**Figure 2 molecules-23-03086-f002:**
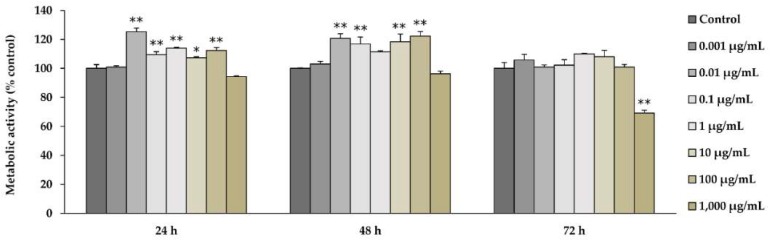
The effect of l-quebrachitol on the cell viability of MC3T3-E1 cells. MC3T3-E1 cells were treated with l-quebrachitol at various concentrations for 24, 48, and 72 h. Cell viability was measured by MTT assay. A representative example of 3 independent experiments. Each data point represents the means of 4 replicate samples ± SEM. * *p* < 0.05 and ** *p* < 0.01 when compared with the control.

**Figure 3 molecules-23-03086-f003:**
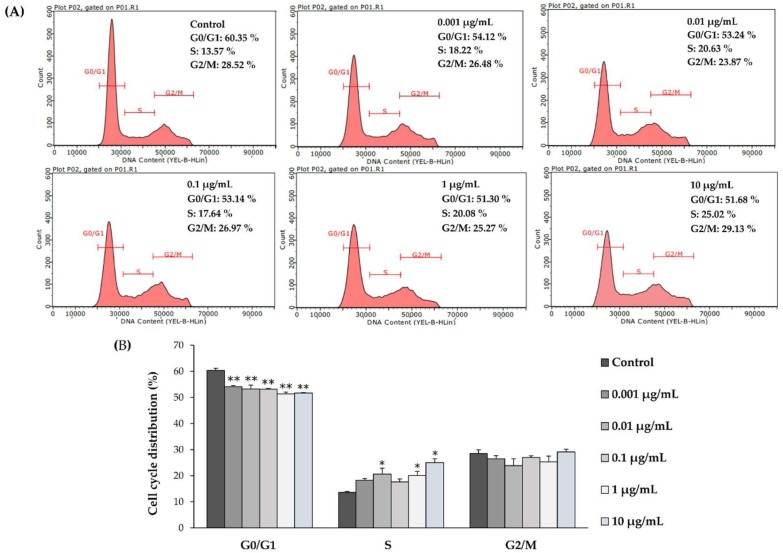
(**A**) The flow cytometry analysis of MC3T3-E1 cells after treatment with l-quebrachitol at concentrations of 0.001, 0.01, 0.1, 1, and 10 µg/mL for 48 h (**B**). The percentage of the total cell population at each phase of the cell cycle is represented by a bar diagram. A representative example of 3 independent experiments. Each data point represents the means of 3 replicate samples ± SEM. * *p* < 0.05 and ** *p* < 0.01 when compared with the control.

**Figure 4 molecules-23-03086-f004:**
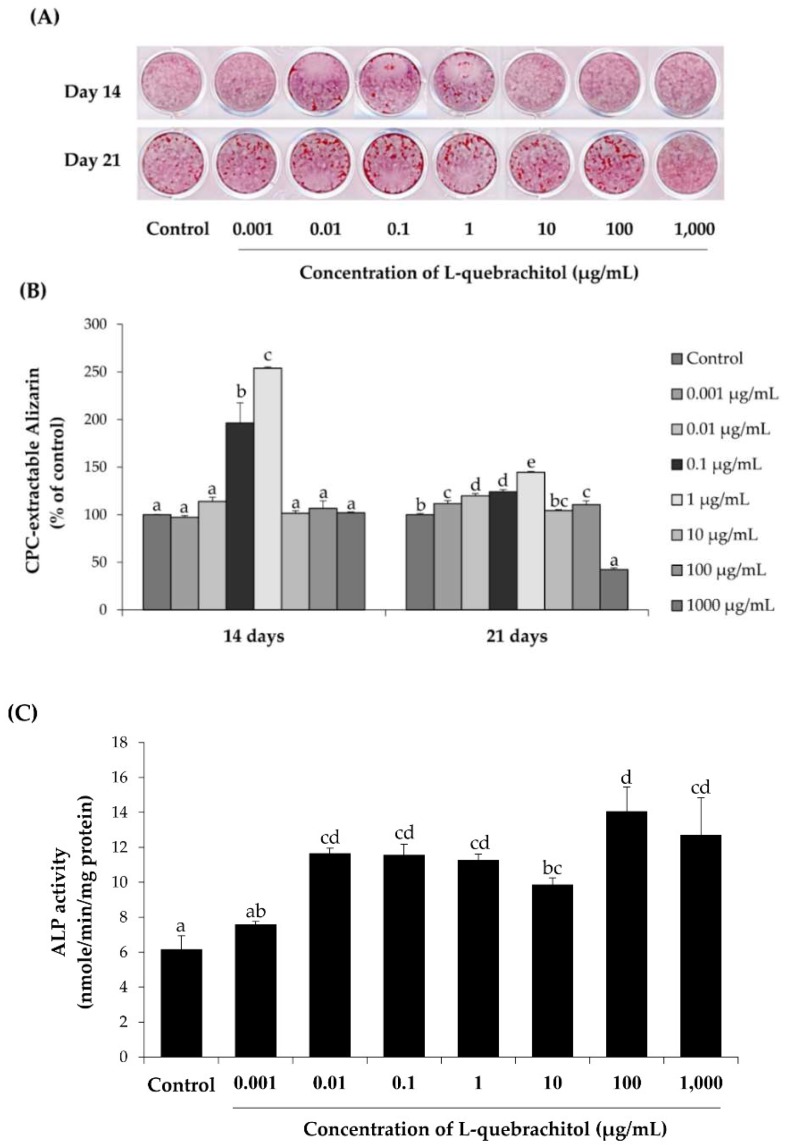
The effect of l-quebrachitol at the concentration of 0.001, 0.01, 0.1, 1, 10, 100, and 1000 µg/mL on osteogenic differentiation and mineralization in MC3T3-E1 cells. (**A**) MC3T3-E1 cells were treated with l-quebrachitol for 14 and 21 days. The cells were stained with Alizarin Red S to observe the production of mineralization (**B**) Quantification of Alizarin Red S staining extracted by cetylpyridinium chloride. (**C**) Alkaline phosphatase (ALP) activity in MC3T3-E1 cells after treatment with l-quebrachitol for 7 days. A representative example of the 3 independent experiments. Each data point represents the means of 4 replicate samples ± SEM. The data in columns with different letters in each group were significantly different at *p* < 0.05.

**Figure 5 molecules-23-03086-f005:**
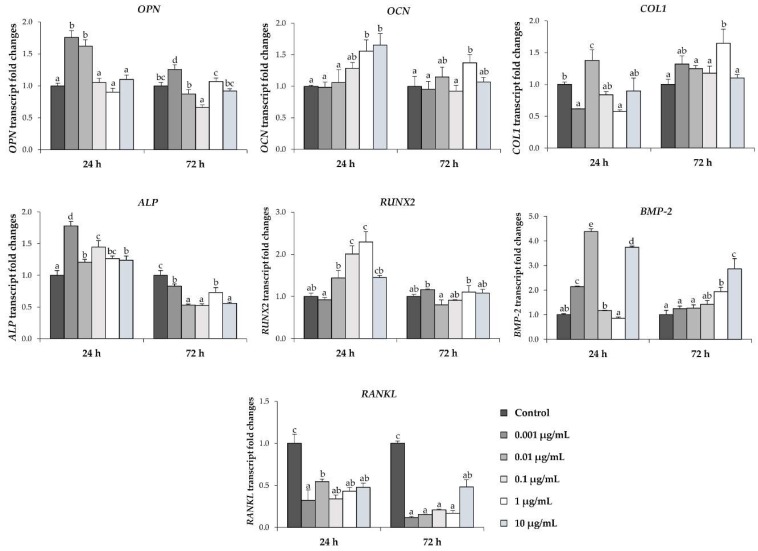
Quantitative real-time PCR was used to analyze the expression of *OPN*, *OCN*, *COL1*, *ALP*, *RUNX2*, *BMP2*, and *RANKL* in osteoblast differentiation from MC3T3-E1 cells treated with the various concentrations of l-quebrachitol for 24 and 72 h. A representative example of 3 independent experiments. Each data point represents the means of 4 replicate samples ± SEM. The data in columns with different letters in each group were significantly different at *p* < 0.05.

**Figure 6 molecules-23-03086-f006:**
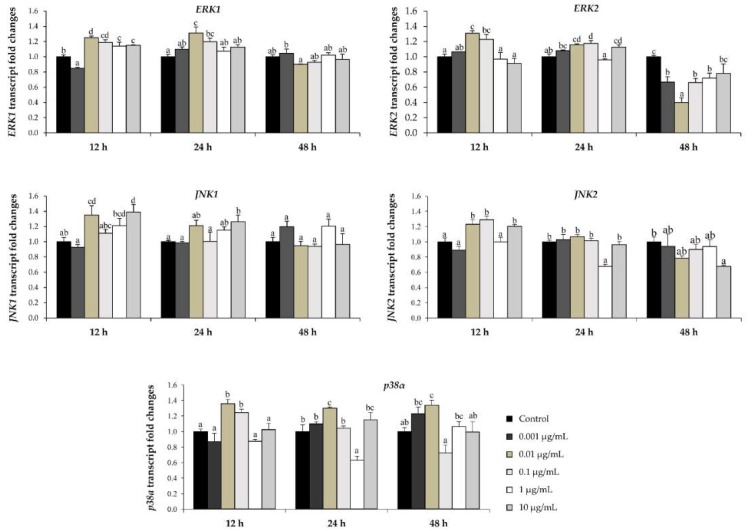
Quantitative real-time PCR was used to analyze the expression of MAPK pathway genes *ERK1*, *ERK2*, *JNK1*, *JNK2*, and *p38α* in osteoblast cells cultured with various concentrations of l-quebrachitol for 24, 48 and 72 h. A representative example of 3 independent experiments. Each data point represents the means of 4 replicate samples ± SEM. The data in columns with different letters in each group were significantly different at *p* < 0.05.

**Figure 7 molecules-23-03086-f007:**
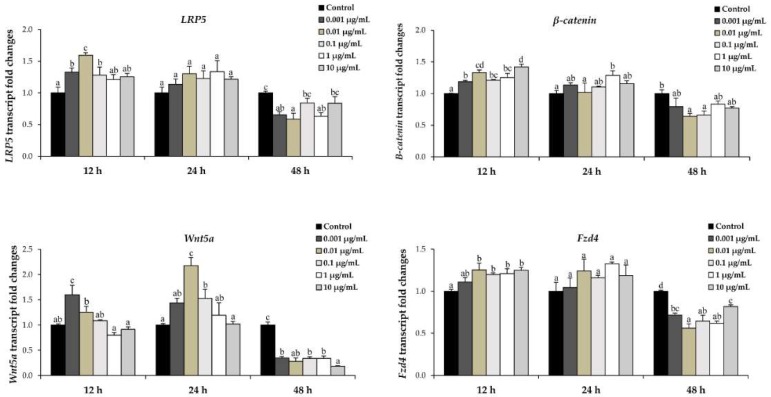
Quantitative real-time PCR was used to analyze the expression of Wnt/β-catenin signaling pathway genes *LRP5*, *β-catenin*, *Wnt5a*, and *Fzd4* in osteoblast cells cultured with various concentrations of l-quebrachitol for 24, 48, and 72 h. A representative example of 3 independent experiments. Each data point represents the means of 4 replicate samples ± SEM. The data in columns with different letters in each group were significantly different at *p* < 0.05.

**Figure 8 molecules-23-03086-f008:**
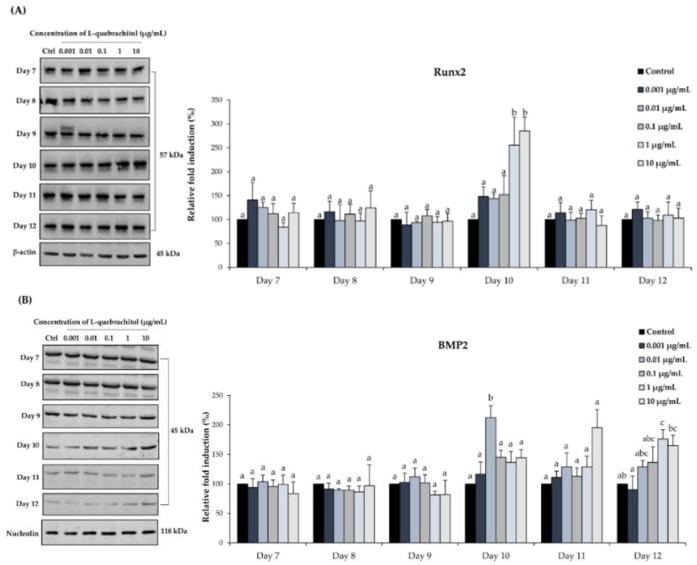
The effects of l-quebrachitol on the protein level expression of Runx2 (**A**) and bone morphogenic protein 2 (BMP-2) (**B**). MC3T3-E1 cells were treated with l-quebrachitol at various concentrations for 7, 8, 9, 10, 11, and 12 days. Levels of each protein(A,B, left-hand panels) were determined by Western blotting using cognate antibodies. Quantitation was performed as described in Materials and methods and levels expressed relative to the loading control proteins β-actin (**A**) and nucleolin (**B**). A representative example of 3 independent experiments. Each data point represents the means of 3 replicate samples ± SEM. The data in columns with different letters in each group were significantly different at *p* < 0.05.

**Table 1 molecules-23-03086-t001:** The primer sequences of qRT-PCR.

Gene	Sequence	GenBank Accession No.
*RUNX2*	F: GTGGCAGTGTCATCATCTGAAAT R: TCGCCTCAGTGATTTAGGGCGCA	NM_001145920.2
*OPN*	F: GCTATCACCTCGGCCGTTGGGG R: CATTGCCTCCTCCCTCCCGGTG	NM_001204203.1
*ALP*	F: ATGGAGGATTCCAGATACAGG R: CCATGGTAGATTACGCTCACA	NM_007431.3
*OCN*	F: CTGTGACATCCATACTTGCAGG R: TGCGCTCTGTCTCTCTGACC	NM_001032298.3
*BMP-2*	F: GCTTCCGCTGTTTGTGTTTG R: GGTCACAGATAAGGCCATTGC	NM_007553.3
*COLI*	F: GCCTTTCCAGGTTCTCCAGCGG R: TTCCCTGGTGCTGATGGTGTTGCT	NM_007742.4
*ERK1*	F: AAGCAGAGACCCCAGCAAAGTGAGAGAAG R: GACACCCCTGTCCTTTTGGATCTGGTCCTG	NM_011952.2
*ERK2*	F: GTATGGGTGGGCCAGAGCCTGTTCAACTTC R: GGTGCCATGGAACAGGTTGTTCCCAAATGC	NM_001357115.1
*JNK1*	F: GCATGGGTCTGATTCTGAAATG R: CTCAGGAGCTCAAGGAATAGTG	NM_001310453.1
*JNK2*	F: AACTCTGCGGATGGTGTTC R: GGTCCTCCATAAAGTCCTGTTC	NM_001163671.1
*p38α*	F: AGGCCATGGTGCATGTGTGT R: AGTAGCTGGAGGAGGAGGAG	NM_001357724.1
*LRP5*	F: TTGACCTTGTGGACCCTTTC R: GAGGACAAGCTCCCACATATT	NM_008513.3
*β-catenin*	F: TCAAGTGAAACCGGGCTATC R: CTCCAACGGGCATCTTCATTA	NM_007664.5
*Wnt5a*	F: ATATCAGGCACCATTAAACCA R: CACTTAGGGGTTGTTCTCTGA	NM_009524.4
*Fzd4*	F: GCACATTGGCACATAAACCGAAC R: GGCTACAACGTGACCAAGATGCC	NM_008055.4
*GAPDH*	F: GAATTTGCCGTGAGTGGAGT R: AAATGGTGAAGGTCGGTGTG	NM_001289726.1

**Notes**: *RUNX2* = runt-related transcription factor 2; *OPN* = osteopontin; *ALP* = alkaline phosphatase; *OCN* = osteocalcin; *BMP-2* = bone morphogenetic protein 2; *COLI* = collagen type 1; *ERK1* = extracellular signal-regulated kinases 1; *ERK2* = extracellular signal-regulated kinases 2; *JNK1* = jun n-terminal kinase 1; *JNK2* = jun n-terminal kinase 2; *LRP5* = low-density lipoprotein receptor-related protein 5; *p38**α* = p38 mitogen-activated protein kinases α; *β-catenin* = Beta-catenin; *Wnt5a* = wingless-type MMTV integration site family, member 5A; *Fzd4* = Frizzled 4 and *GAPDH* = glyceraldehyde-3-phosphate dehydrogenase.

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
