# Peer review of "l-Quebrachitol Promotes the Proliferation, Differentiation, and Mineralization of MC3T3-E1 Cells: Involvement of the BMP-2/Runx2/MAPK/Wnt/β-Catenin Signaling Pathway"

_molecules, 2018, doi:10.3390/molecules23123086_

Round 1

Reviewer 1 Report

The manuscript is very well written and the results are conclusive.

With respect to the putative proproliferative effects on MC3T3 cells, it would be helpful to explore whether this effect is cell specific or if other (primary) cell types might be affected as well. Are there any effects on cancer cell lines?

Author Response

We agree with the referee that the effects of L-Quebrachitol on cell proliferation is of interest. However, given that the focus of the current work was to explore the effects of L-Quebrachitol on osteoblastogenesis, it was important to focus on a well-established model of osteoblastogenesis, such as the widely used MC3T3-E1 cells utilised in this study.  The effects on other primary cell types or cancer cells, while interesting in their own right, would not ADD to the conclusions found here, that this particular polyol isomer promotes differentiation and mineralisation as part of the osteoblastogenic pathway in an appropriate model cell system.

Reviewer 2 Report

Comments

1.dose of L-quebrachitol

According to Fig.2, less than 10µg/ml L-quebrachitol does not significantly influence cell viability, the question is following;

Have authors detected IC50 or pIC50 of the compound?  

2. Cell passages

 Chung CY, et al  showed cell passages of  MC3T3-E1  is closely associated with genes expression (Serial passage of MC3T3-E1 cells alters osteoblastic function and responsiveness to transforming growth factor-beta1 and bone morphogenetic protein-2. Biochem Biophys Res Commun. 1999 Nov;265(1):246-51). What about cell passages details in this manuscript?   Effects of Longer culture time on gene expressed should be considered.

3 Cell proliferation assay

Either Br-dU or E-dU should be used for cell proliferation assay, and MTT just reflects cell viability. Please adjust title.

4. Fig.6 and Fig.7

Protein level assay, such as western blot, should be used here. Please add related results.

5. Language style

Too many grammar structural errors and typos could be found in the current manuscript. Authors should use scientific description, and professional edit is necessary.

Author Response

Point 1: dose of L-quebrachitol

According to Fig.2, less than 10µg/ml L-quebrachitol does not significantly influence cell viability, the question is following;

Have authors detected IC50 or pIC50 of the compound?  

Response 1: We are very grateful to this reviewer for their insightful suggestions. An IC50 value has NOT been determined for this compound using these cells. As can be seen from the 72h timepoint, L-quebrachitol at a concentration of 1 mg/ml does reduce viability somewhat. This is a very high concentration of such a compound, and as such reflects the remarkable tolerance cells have for this type of compound. Practical considerations regarding the concentration and properties of stock solutions required to determine an IC50 preclude the determination of IC50. As the primary purpose of the work was to investigate the role of L-quebrachitol in osteoblastogenesis, we do not feel that a determination of an IC50, which is likely to be a concentration far above that ever used in model studies or any potential therapeutic setting, will add significant value to the manuscript.

Point 2: Cell passages

 Chung CY, et al showed cell passages of  MC3T3-E1  is closely associated with genes expression (Serial passage of MC3T3-E1 cells alters osteoblastic function and responsiveness to transforming growth factor-beta1 and bone morphogenetic protein-2. Biochem Biophys Res Commun. 1999 Nov;265(1):246-51). What about cell passages details in this manuscript?   Effects of Longer culture time on gene expressed should be considered.

Response 2: We are grateful to the reviewer for pointing this out. We have indicated in the study the passage numbers of the MC3T3-E1  cells used in various experiments, and we have incorporated the reference into the bibliography.

Point 3: Cell proliferation assay

Either Br-dU or E-dU should be used for cell proliferation assay, and MTT just reflects cell viability. Please adjust title.

Response 3:We have adjusted the manuscript in various places to more correctly reflect the significance of the MTT assay and the results obtained from it. We disagree with the reviewer in suggesting that MTT simply reflects cell viability alone, as it is in fact a measure of the total metabolic activity in a particular sample, and as such will be a proxy indicator of cell number. We have changed the annotation of the Y axis in Figure 2 to reflect this. Compounds that block cell proliferation over an extended period will show reduced MTT signal compared with controls WITHOUT necessarily having any impact on the viability of cells (this is further discussed in Ramu et al, Ramu, V., Gill, M. R., Jarman, P. J., Turton, D., Thomas, J. A., Das, A., & Smythe, C. (2015). A Cytostatic Ruthenium(II)-Platinum(II) Bis(terpyridyl) Anticancer Complex That Blocks Entry into S Phase by Up-regulating p27(KIP1). Chemistry - a European Journal, 21(25), 9185–9197. http://doi.org/10.1002/chem.201500561)

The referee suggests using either BrdU or EdU for analysis of cell cycle profiles of cells treated with L-quebrachitol. We disagree with the referee that the use of bivariate FACS analysis would add any further useful information to the data already acquired. The acquisition of estimates of cells in each of the phases of the cell cycle using univariate analysis of DNA content is very well established (Watson, Chambers, & Smith. A pragmatic approach to the analysis of DNA histograms with a definable G1 peak. Cytometry 8:1-8 (1987)). While it is true that univariate FACS analysis MAY underestimate the proportion of cells in S phase under certain special circumstances (such as when cells are subjected to replication stress – discussed in Feijoo, C. (2001). Activation of mammalian Chk1 during DNA replication arrest: a role for Chk1 in the intra-S phase checkpoint monitoring replication origin firing. The Journal of Cell Biology, 154(5), 913–924. http://doi.org/10.1083/jcb.200104099), the extent of under-estimate is consistent across all samples  treated in the experiments presented here  as identical gating conditions were applied for the determination of the S-phase fraction in the presence of varying concentrations of L-quebrachitol.

Point 4: Fig.6 and Fig.7

Protein level assay, such as western blot, should be used here. Please add related results.

Response 4: The purpose of the experiments presented in Fig 6 and 7 were to demonstrate that L-quebrachitol induces changes in gene expression which are consistent with well-known and extremely well-characterised cell signalling pathways. The changes are clearly demonstrated. While we agree that it MIGHT be interesting to confirm that such gene expression changes result in alterations in their cognate protein levels, this level of analysis is beyond the scope of this manuscript, and a result, either way, would not materially alter the conclusions drawn.

Point 5: Language style

Too many grammar structural errors and typos could be found in the current manuscript. Authors should use scientific description, and professional edit is necessary.

Response 5: We are grateful to the referee for this advice, which has been undertaken.

Reviewer 3 Report

This manuscript needs to be edited line-by-line before it can be properly reviewed.  There are numerous grammatical errors:  Subjects and verbs do not agree with each other.  Verb tenses are used in a haphazard (and often incorrect) way.  The definite article is used when it should not be used and vice versa.  Singular and plural are mixed up.  There are run-on sentences and incomplete sentences.  Last but not least, the writing is very verbose.  It rarely says with one word what can be said with three or four words.  For a typical example see lines 258-259:  "...latex serum has potential as a source for utilization and application of a wide range of biologically active constituents and enzymes.."  How about "..latex serum has potential as a source of biologically active compounds"? 

Far too many sentences start with "It has been observed that" or "It has been demonstrated.that."  It is sometimes OK to simply say "the grass is green".  There is no need to resort to "it has been demonstrated that the grass is green".

OTHER ISSUES

• What are we supposed to see in Figure 4A?  I cannot see any major differences between the individual dishes.

• "ALP" should be defined at the time of first use.

• Lines 133-134:  If the recovery yields of 0.864 g/L and 1.728 g/L, respectively represent average yields, standard deviations should be provided.  If the experiment was done only once, this should be clearly stated.  If this manuscript reports the result of a single experiment, results should be soft-pedaled, as reproducibility has not been established.  By the way, the word "about" looks out of place next to such precise recovery yields. 

• When referring to sample sizes (e.g. n=4 in legend to Figure 4), do you refer to 4 replicate culture dishes for each data point of a single experiment or to 4 independent experiments?  If you refer to 4 independent experiments, how many replicate dishes were set up per data point. 

• Figure 2, label of y-axis:  Viabilities in excess of 100% do not make sense.  They question the appropriateness of the MTT assay in this context. 

Figure 4B:  "Absorbance of Alizarin solubility" sounds awkward.  How about "CPC-extractable Alizarin (% of control)"?

• When indicating the source of reagents, name AND location of the supplier should be provided.

Author Response

Point 1: This manuscript needs to be edited line-by-line before it can be properly reviewed.  There are numerous grammatical errors:  Subjects and verbs do not agree with each other.  Verb tenses are used in a haphazard (and often incorrect) way.  The definite article is used when it should not be used and vice versa.  Singular and plural are mixed up.  There are run-on sentences and incomplete sentences.  Last but not least, the writing is very verbose.  It rarely says with one word what can be said with three or four words.  For a typical example see lines 258-259:  "...latex serum has potential as a source for utilization and application of a wide range of biologically active constituents and enzymes.."  How about "..latex serum has potential as a source of biologically active compounds"? 

Far too many sentences start with "It has been observed that" or "It has been demonstrated.that."  It is sometimes OK to simply say "the grass is green".  There is no need to resort to "it has been demonstrated that the grass is green".

Response 1: We are grateful to the referee for this advice, which has been undertaken. The manuscript has been revised and edited throughout.

OTHER ISSUES

Point 2: What are we supposed to see in Figure 4A?  I cannot see any major differences between the individual dishes.

Response 2: We have reconstructed Figure 4 and enlarged the relevant section so that the image of cells stained in situ with Alizarin red is more obvious. The images of cells show that the induction of mineralization is not uniform in the cell populations. While the reasons for this are currently unknown, it is likely related to heterogeneity in the precise timing of differentiation within individual cells. The data in Figure 4B provide quantitative values that correspond to the visual representations in Figure 4A, however, to not present evidence of the observed variation in a cell population undergoing mineralization would be misleading to the readership. 

Point 3: "ALP" should be defined at the time of first use.

Response 3: This has now been done.

Point 4: Lines 133-134:  If the recovery yields of 0.864 g/L and 1.728 g/L, respectively represent average yields, standard deviations should be provided.  If the experiment was done only once, this should be clearly stated.  If this manuscript reports the result of a single experiment, results should be soft-pedaled, as reproducibility has not been established.  By the way, the word "about" looks out of place next to such precise recovery yields. 

Response 4: Indicators of statistical variation in recovery have been added and the text revised to reflect this.

Point 5: When referring to sample sizes (e.g. n=4 in legend to Figure 4), do you refer to 4 replicate culture dishes for each data point of a single experiment or to 4 independent experiments?  If you refer to 4 independent experiments, how many replicate dishes were set up per data point. 

Response 5: This has been clarified in the Figure legends in each case.

Point 6: Figure 2, label of y-axis:  Viabilities in excess of 100% do not make sense.  They question the appropriateness of the MTT assay in this context. 

Response 6: The Y axis has been changed to more accurately reflect the output of the MTT assay.  Please see response to Reviewer 1.

Point 7: Figure 4B:  "Absorbance of Alizarin solubility" sounds awkward.  How about "CPC-extractable Alizarin (% of control)"?

Response 7: This has been changed as requested.

Point 8: When indicating the source of reagents, name AND location of the supplier should be provided.

Response 8: This has been changed as requested.

Round 2

Reviewer 3 Report

The legends to Fig. 2-8 should clearly indicate if the figures show representative examples of independent experiments that were repeated multiple times, or if they show the results of experiments that were performed only once.  For example:  "Effects of L-quebrachitol on the....... Representative example of 5 independent experiments.  Each data point represents the means of 3 replicate samples ± SEM." 

Along the same line:  Were single batches of latex and serum used for the entire study?  If multiple batches were used, yields should be presented as means ± SEM, and the number of batches should be indicated (n=....).

Author Response

This has been clarified in the Figure legends in each case and the number of batches has been indicated as requested.
